# Skull R-CNN: A CNN-based network for the skull fracture detection

**Zhuo Kuang**[1]                                                    kuangzhuo@hust.edu.cn
**Xianbo Deng**[2]                                                  dengxianbo@hotmail.com
**Li Yu**[1]                                                        hustlyu@hust.edu.cn
**Hang Zhang**[2]                                                   790731431@qq.com
**Xian Lin**[1]                                                     linxian@mails.ccnu.edu.cn
**Hui Ma**[2]                                                       1363191284@qq.com

[1] *Huazhong University of Science and Technology, China*

[2] *Union Hospital Affiliated with Tongji Medical College of Huazhong University of Science and Technology, China*

## Abstract

Skull fractures, following head trauma, may bring several complications and cause epidural hematomas. Therefore, it is of great significance to locate the fracture in time. However, the manual detection is time-consuming and laborious, and the previous studies for the automatic detection could not achieve the accuracy and robustness for clinical application. In this work, based on the Faster R-CNN, we propose a novel method for more accurate skull fracture detection results, and we name it as the Skull R-CNN. Guiding by the morphological features of the skull, a skeleton-based region proposal method is proposed to make candidate boxes more concentrated in key regions and reduced invalid boxes. With this advantage, the region proposal network in Faster R-CNN is removed for less computation. On the other hand, a novel full resolution feature network is constructed to obtain more precise features to make the model more snesetive to small objects. Experiment results showed that most of skull fractures could be detected correctly by the proposed method in a short time. Compared to the previous works on the skull fracture detection, Skull R-CNN significantly reduces the false positives, and keeps a high sensitivity.

**Keywords:** Convolutional neural networks, Skull fracture, Object detection, Skeletonization

## 1. Introduction

The presence of a skull fracture following head trauma can lead to several complications, and increase the risk of an underlying subdural or epidural hematoma (de Boussard et al., 2006). Therefore, the detection of skull fractures is very important for the evaluation of traumatic head injury.

Nowadays, the cranial computed tomography (CT) has become the diagnostic standard of care for suspected skull or brain injury (for Acute Care , UK et al.(2007). The radiologist can diagnose the presence of skull fractures based on CT. But, as shown in Fig.1, on CT scans, the skull fracture presents the following characteristics: (1) The fractures usually present as narrow slits; (2) The locations and the length of fractures are diverse; (3) A considerable percentage of the fractures have very small sizes. All these features will make

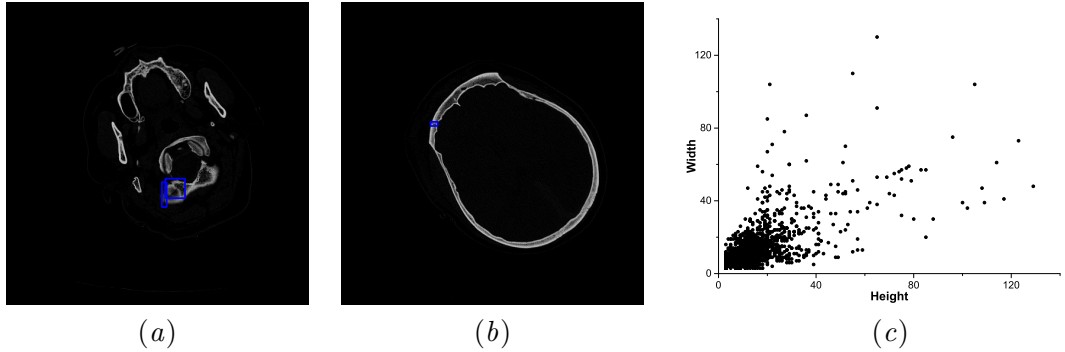

$(a)$ $(b)$ $(c)$

Figure 1: (a-b) The skull fractures annotated by the radiologist. The blue boxes are the ground truth annotated by the radiologists, which contain the fractures. (c) The distribution of the width and length of the object boxes.

the manual detection time-consuming and laborious. Thus, presenting an efficient automatic skull fracture detecting method is very significant.

Some methods have been already proposed for the skull fracture detecting. H.Shao et al. (Shao and Zhao, 2003) proposed region growing method to segment the brain and then used the entropy function to generate the rules for the skull fracture diagnosis. Wan et al. (Zaki et al., 2009) applied Sobel edge detection technique for the identification of skull fracture. Ayum et al. (Yamada et al., 2018) proposed a surface selective Black-Hat transform method for the skull fracture detection, and reached a sensetivity of 86%. However, all the methods mentioned above only considered the local features of the fractures, and there were many false positives among the prediction results.

Recently, deep learning models, especially the convolutional neural networks (CNNs) have been widely applied into the image processing. Compare to traditional methods, CNNs can extract the high-dimensional features of the image and have shown excellent performance in the object detection, image classification and segmentation tasks (Ronneberger et al., 2015; Ren et al., 2015; Liu et al., 2016; He et al., 2017). In the area of object detection, the CNN-based methods can be divided into two kinds: the region-proposal networks, like Faster R-CNN (Ren et al., 2015); the region-free networks, like SSD (Liu et al., 2016) and YOLOV3 (Redmon and Farhadi, 2018). Compared to the latter kind, Faster R-CNN takes more time on detecting, but obtains more accurate results especially for the small objects. Although Faster R-CNN have been proved that it could well detect the objects in natural images, but there are some problems when it is applied into the skull fracture detecting directly. First, lots of the ground truth boxes of the fractures have very small size, thus, suitable candidate regions are difficult to be obtained. Moreover, the time consumption may be very large, because for one CT scan, there are more than a hundred of 2D slices.

In this paper, aiming at the skull fracture detection, we propose a novel CNN-based model, named as Skull R-CNN. Guiding by the morphological features of the skull, a skeleton-based region proposal method is designed for more reasonable candidate regions. On the other hand, a novel full resolution feature network is constructed, which can optimize the feature extraction for the small objects. The experiments results show that Skull

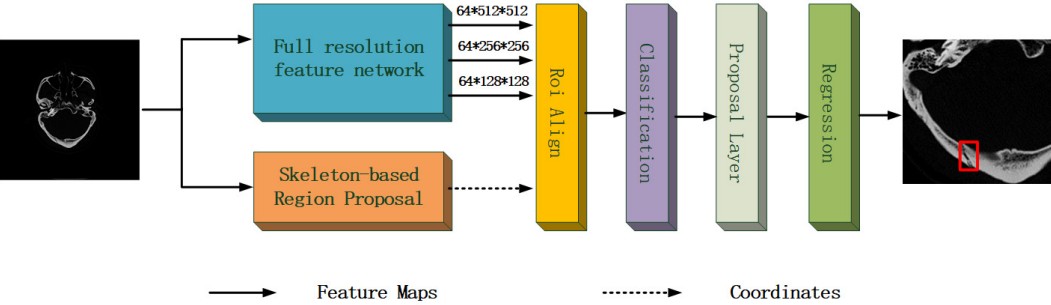

Figure 2: The architecture of the Skull R-CNN.

R-CNN achieve the AP of 0.65 on the validation set, 0.60 on the test set and reduce time consumption by 60% than Faster R-CNN. Compared the previous works on the skull fracture detecion, Skull R-CNN significantly reduces the less false positives, and keep a high sensitivity.

## 2. Skull R-CNN

The proposed method integrates the skull morphological features with deep convolutional networks to accurately detect the skull fractures. Fig.2 shows the pipeline of our proposed Skull R-CNN. First, multi-scale feature maps and all candidate region boxes are obtained, then with the help of the Roi-Align (He et al., 2017), the corresponding feature maps of each box is adjusted to the same dimension. Next, the classification module will give a score to each candidate box and screen out suitable boxes. Finally, the position of the boxes will be fined tune to better contain the fractures.

### 2.1. Skeleton-based region proposal

In Faster R-CNN, the number of the candidate boxes can be calculated as following:

$$N = K \times w_f \times h_f \tag{1}$$

Where $h_f$, $w_f$ represent the length and width of the feature map. Every point on the feature map is considered as an anchor, and with the different sizes and different length-width ratio, K boxes are built around the anchor. It is obvious that, the density of the boxes depends on the downsampling times and stride. With the increase of downsampling times and step length, the boxes become more and more sparse, and the sparse boxes cannot contain the small objects well. On the other hand, by this way, the candidate boxes are evenly distributed on the whole image, this will result in a lot of computational redundancy. Especially for the skull fracture detecting, the fractures exist only on the skull, not the rest of the CT image.

In this work, we put forward a skeleton-based region proposal method. Different from the Faster R-CNN, anchors are generated on the origin image directly. As shown in Fig.3, we first apply a skeletonization method (Lam et al., 1992) to obtain the skull skeletons $S_s$. Then the whole image is divided into $256 \times 256$ grids, the gridlines are defined as $S_g$,

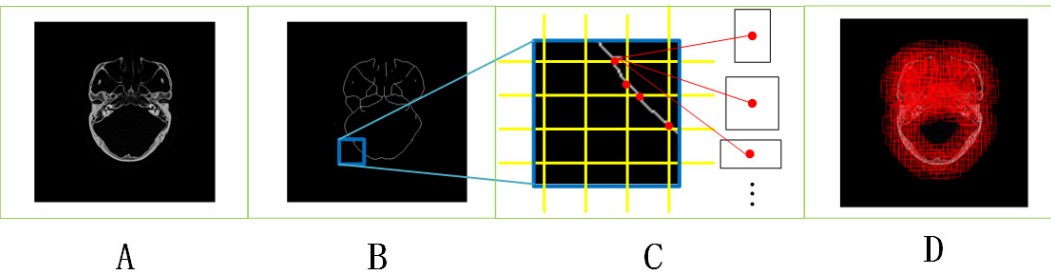

Figure 3: The skeleton-based region proposal method. A is the origin image; B is the skeleton. C is the anchors and candidate boxes in a part of the image.D is the final results of the candidate boxes.

anchors are generated by the following:

$$A_c = S_s \cap S_g \tag{2}$$

Finally, the sizes of 8, 16, 32 and length-width ratio of 0.5, 1, 2 are applied to build the boxes around the anchors. According to this method, the number of the candidate boxes varies with the shape of the skull. For the simple shape, the boxes will be fewer, and for the complex shape, there will be more boxes to contain the skull. In any case, these boxes will be very densely distributed on the whole skull. With these dense boxes, there is no need to apply the region proposal network (RPN) (Ren et al., 2015) for the rough selection and offset.

### 2.2. Full resolution feature network

In the feature pyramid network (FPN) (Lin et al., 2017), the application of multi-scale features has been proved to improve the performance of detecting objects with different sizes. In which, the smaller objects are detecting by the feature maps with higher resolution for more local information. However, the highest resolution of the output feature maps in FPN is only a quarter of the origin image, for the small skull fractures, it is hard to be effective.

To obtain the more appropriate feature maps for the skull fracture detecting, we introduce the encoder-decoder structure. Networks with this structure have been widely applied into the segmentation field, in which U-Net (Ronneberger et al., 2015) has achieved a high accuracy on many tasks. Inspired by U-Net, we construct a novel full resolution feature network. As Fig.4 shows, the backbone is similar to the U-Net, 3 times of downsampling and upsampling with the skip connection. Different from the element-wise addition in FPN, concatenation is applied here to merge the feature maps, making the weight between the local features and high-level semantic features learnable. To obtain the feature maps of full, quater, and eighth resolution, 1*1 convolutional layer is used to reduce channel dimensions of the last three upsampling results, and make them to the same.

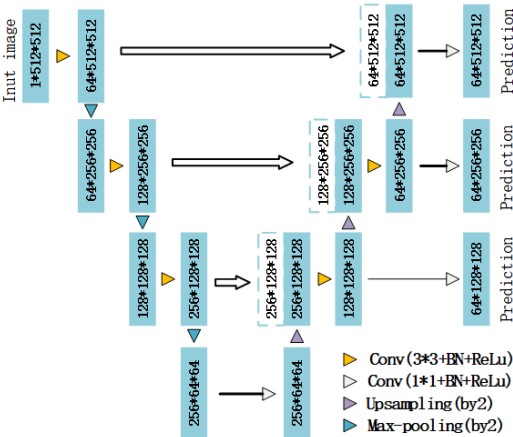

Figure 4: The structure of the full resolution feature network.

### 2.3. Classification and regression

Because the candidate boxes are generated on the origin image, coordinates of these boxes mapped on the feature maps may not be integers. To get more accurate features for every candidate boxes, we choose the ROI-Align proposed in Mask R-CNN(He et al., 2017) to replace the ROI-Pooling. Then a classification network with fully connected layers scores all the boxes within 0 to 1 (binary classification), and the Non-Maximum Suppression (NMS) is adopted to get the boxes with the highest score in the local area. Finally, with a regression network, the location offsets of these selected boxes are obtained. The form of offsets is the same as Faster R-CNN, the offsets of the regression center and the length-width scaling of the box ($\Delta_1 \sim \Delta_4$). The final coordinates of the predicted box can be expressed as follows:

$$x'_c = x_c + \Delta_2 \times w; y'_c = y_c + \Delta_2 \times h \tag{3}$$

$$w' = w \times e^{\Delta_3}; h' = h \times e^{\Delta_4} \tag{4}$$

Where $x_c, y_c, w, h$ are the center coordinates and width and height of the box, $x_c$ and $x'_c$ are for the candidate box and the predicted box (likewise for $y, w, h$). In this form, the range of the output could be limited and the training process could be sped up.

### 2.4. Training

As mentioned above, the accuracy of classification is the key to decide the performance of the proposed Skull R-CNN. To train the classification module, taking the ground truth as standard, the labels of the candidate boxes are defined as below:

$$label = \begin{cases} 0 & \text{IOU} < 0.3 \\ 1 & \text{IOU} \geq 0.5 \end{cases} \tag{5}$$

However, the ratio between positive and negative samples is extremely unbalanced, negative samples are far more than the positive samples. To solve this problem, in the training

Table 1: The value of offsets. $\delta_x$ represents the $x$ coordinates and $\delta_y$ represents the $y$ coordinates.

| $\delta_x(\times\frac{w}{5})$ | $0,0$ | $-1,0$ | $+1,0$ | $0,-1$ | $0,+1$ | $-1,-1$ | $+1,+1$ |
|---|---|---|---|---|---|---|---|
| $\delta_y(\times\frac{h}{5})$ | $0,0$ | $-1,0$ | $+1,0$ | $0,-1$ | $0,+1$ | $-1,-1$ | $+1,+1$ |

process, some extra artificial positive sample are introduced. For each ground truth box ($GT$), the extra samples ($P_e$) can be expressed as follows:

$$P_e = \{P_e^{i,j}|GT + floor([\delta_x^i, \delta_y^j])\} \tag{6}$$

Where $\delta_x$ and $\delta_y$ are offsets for the two corners, and as shown in Table.1, they both have 7 groups of values, in which $w$ and $h$ are the width and length of the ground truth. In total, 48 extra positive samples are obtained, then the negative samples for training are randomly selected by the ratio of 2:1 to the positive samples.

The training process is divided into two steps. Firstly, we train the backbone and the classification module, with Binary Cross Entropy (BCE) as the loss function. Then, based on the weight obtained in the first step, the whole model is trained, and smoothL1 is set as the loss function for the regression module. In the second step, to balance the two kinds of losses, the weighted loss is defined as following:

$$L = L_c + 10 \times L_r \tag{7}$$

In which, the $L_c$ is the loss of the classification task and $L_r$ is the loss of the regression task, weighing $L_r$ by 10 to balance the two losses.

## 3. Experiments and results

### 3.1. Data

In this study, we collected the CT scans from 45 head trauma patients. All the collected CT scans were DICOM format. They were composed of $512 \times 512$ pixels with $1.25mm$ slice thickness. The collected CT scans were annotated by the radiologists with more than 10 years of experience. To keep the diversity among the samples, for each scan, we took one of every two slices as a sample. Totally, 872 slices were obtained for the experiment.

### 3.2. Implementation details

First of all, we transformed the ICH CT scans from DICOM format to gray images. Considered that the CT intensity (HU) of the skull is larger than 1000, the transformational rule was defined as the following:

$$I = \begin{cases} 255 & \text{HU} \geq 2550 \\ \frac{HU}{10} & 0 < \text{HU} < 2550 \\ 0 & \text{HU} < 0 \end{cases} \tag{8}$$

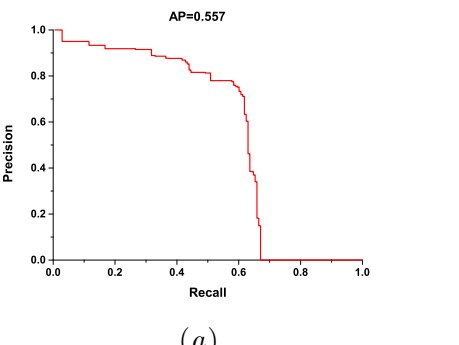 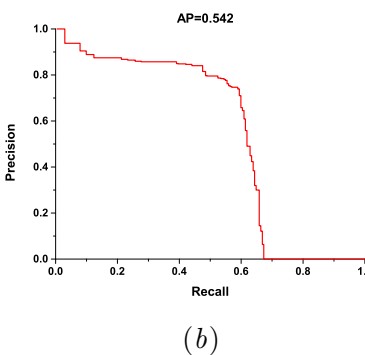

Figure 5: The PR cures of the Faster R-CNN+FPN.(a) the validation set (b) the test set.

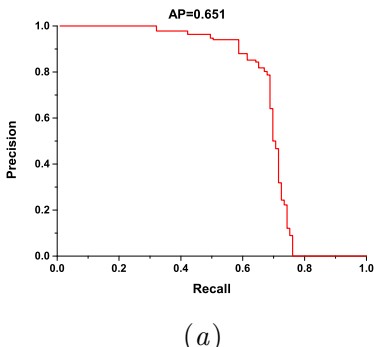 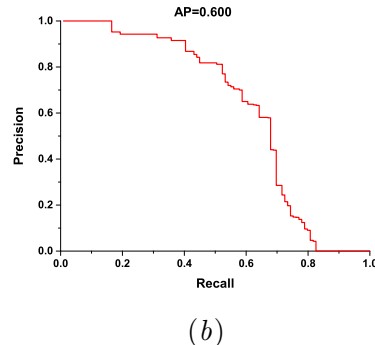

Figure 6: The PR cures of the Skull R-CNN.(a) the validation set (b) the test set.

Then,25 scans with 491 slices were selected as the training set, 10 scans with 208 slices as the validation set, 10 scans with 173 slices as the test set. It is worth to be mentioned that, to verify the performance of the Skull R-CNN on detecting small fracrures, we collected the slices with the ground truth boxes smaller than $16 \times 16$, and 101 slices of the validation set and 93 slices of the test set were obtained. For the both two steps of the training process, RMSprop with $\alpha = 0.9$ was selected as the optimization function. The initial learning rate was set to 0.0001, which was subsequently reduced by a factor of 0.1 for every 20 epochs. All the experiments were finished on a GTX 1080ti graphics card.

### 3.3. Detection results

The average precision (AP) was used to quantitatively evaluate the detection performance. AP is the area under the Precision-Recall (PR) cure, where $precision = \frac{TP}{TP+FP}$, $recall = \frac{TP}{TP+FN}$. The TP, FP, and FN refer to true positives, false positives, and false negatives, respectively. In which, the detection results with the IOU greater than 0.5 were defined as the true object boxes. To make a benchmark for our proposed method, we applied Faster R-CNN to the skull fracture with FPN as the backbone, and the changed the sizes of anchor boxes according to the clustering results. The PR cures for the validation set and the test set are plot in Fig.5 and Fig.6, and the AP values are shown in Table.2.

Table 2: The performance of the models.

| Methods | AP(×0.01) | | | | Detection time(s\slice) | |
|---|---|---|---|---|---|---|
| | val | test | val(<16*16) | test(<16*16) | val | test |
| Faster R-CNN + FPN | 55.7 | 54.2 | 59.4 | 49.3 | 0.088 | 0.087 |
| Skull R-CNN + FPN | 62.6 | 57.9 | 64.7 | 58.6 | 0.058 | 0.058 |
| Skull R-CNN | 65.1 | 60.0 | 67.3 | 63.3 | 0.035 | 0.036 |

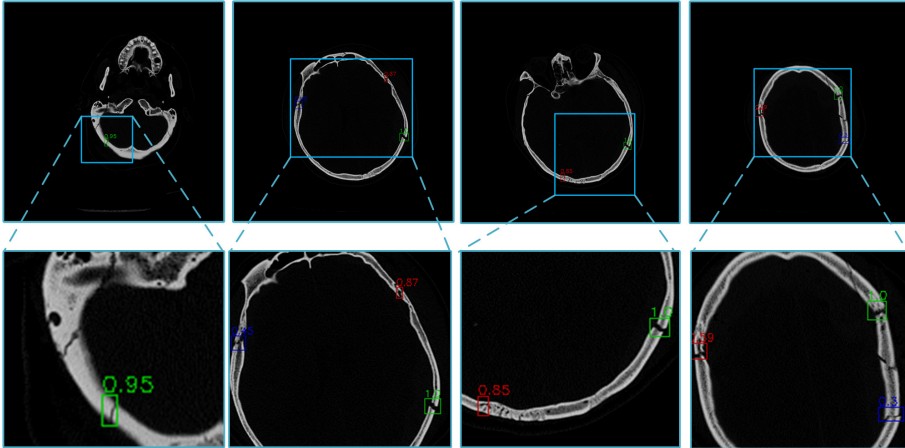

Figure 7: The detection results of the Skull R-CNN. The images in the second row are the partial magnifications of images in the fist row. In which, the green boxes are TP predictions, the red boxes are FP predictions, and the blue boxes are the FN predictions.

On the two sets, Faster R-CNN+FPN gets the best possible recall(BPR) no more than 0.7, while the Skull R-CNN achives average of 0.8. As for the overall performance, compared to Faster R-CNN+FPN, the whole Skull R-CNN improves the AP by an average of 0.075. And the succinct architecture reduces time cost by 60% for detecting. It should be noted that, as for the small objects, the improvements are around 0.11. This demonstrates that the Skull R-CNN could better detect small fractures. Morever, we relpace the full resolution feature network by the FPN and get the combination of Skull R-CNN+FPN. This combination has a performance between the Faster R-CNN+FPN and the whole Skull R-CNN over the four datasets, demonstrating that the full resolution feature network and the skeletonization method both play a important role in the skull fracture detecting.

With the score of 0.5 as the classification threshold, some detection results (true negetive boxes not included) are shown in Fig.7. As a whole, most of the fractures can be detected correctly, and the prediction boxes can contain the fractures well. The biggest constraints on performance is that some sagittal, lambdoid and coronal sutures are also detected as fractures. Besides, sometimes the regression module can not give a precise offsets to the

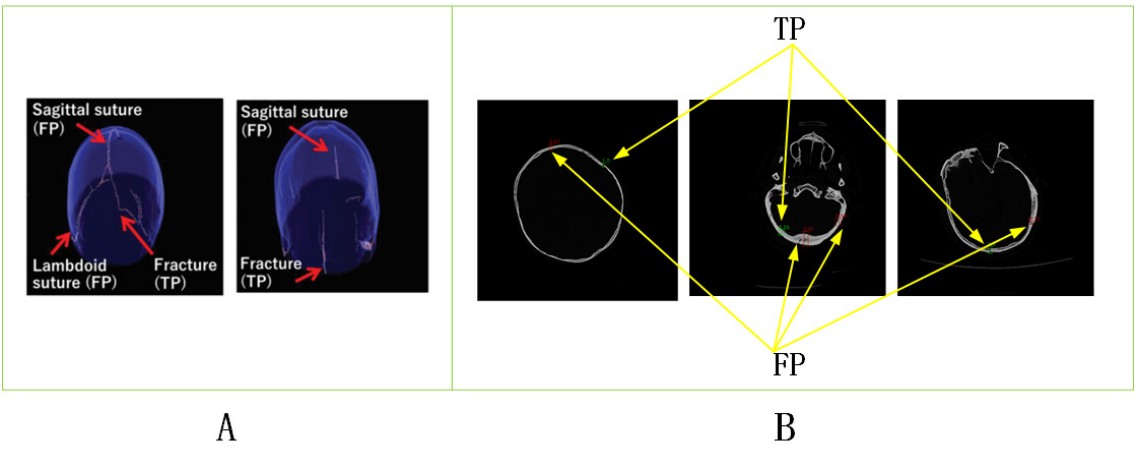

Figure 8: A comparision of the subjective results. A is the results shown in (Yamada et al., 2018), all the white lines are the predicted fractures. B is the detection results predicted by Skull R-CNN.

boxes. For example, in the fourth image of Fig.7, the box with the score of 0.59 contain the fracture actually, but the area is some larger than the ground truth.

To the best of our knowledge, the 3d black-Hat transform reported in (Yamada et al., 2018) achivied the best performance, the sensitivity of 86% with 6.83 FPs per scan. Because our method focus on the 2d CT slices, and the data sets are inconsistent, the objective indicators cannot be unified. A subjective comparison is shown in Fig.8. Expect for the suture lines, many other normal structures, like the vascular grooves, were also detected as false positives by the black-Hat method. On the contrary, parts of the suture lines can be determined by the Skull R-CNN. And the other small number of FNs are scattered on the continous slices, which are easy to be determined by the radiologists.

## 4. Conclusion and discussion

In this paper, based on Faster R-CNN, a novel Skull R-CNN was proposed for the automatic skull fracture detection. In which, a skeleton-based region proposal method was designed to replace the RPN with less computation and more reasonable candidate boxes. In addition, a full resolution network was constructed to extract more precise features for the small objects detection. The proposed Skull R-CNN can accurately detect the proposed method in a short time. Compared to the previous studies on the skull fracture detecion, detection results of Skull R-CNN have less false positives and keep a high sensitivity. Future works will focus on improving the ability of the proposed network to distinguish skull fractures from suture lines.

## Acknowledgements

This work was supported in part by the NSFC under Grant 61871437, in part by the Natural Science Foundation of Hubei Province of China under Grant 2019CFA022.

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
