# OpenReview forum: "Skull-RCNN: A CNN-based network for the skull fracture detection"
_MIDL.io/2020/Conference — MIDL 2020_

### Official Review · AnonReviewer1 · 2020-03-10
**Simple way of including prior knowledge in region proposal based object detection**

**Rating:** 3
**Confidence:** 4

**Summary:**

This paper describes a variant of the Faster R-CNN object detection method for detection of skull fractures in CT scans. Instead of considering region proposals across the entire image, following a regular grid, the paper proposes to run an ad-hoc edge detection and skeletonization method on the image to obtain a very rough segmentation of the skull and to generate region proposals in an unsupervised way by placing an array of differently sized region proposals across the skull. The method works on 2D slices of head CT scans and is quantitatively compared with Faster R-CNN and qualitatively with another skull fracture detection method.

**Strengths:**

The proposed idea is an attempt to include prior knowledge about the structure of interest (bone has high HU values in CT, the skull has clear edges with respect to other structures in the image) in a well-known object detection framework. This is something that makes a lot of sense to me for object detection in medical images, where we typically have such prior knowledge, and where such a simple strategy can avoid implausible false positive detections (at least to some extent).

**Weaknesses:**

The paper is sometimes a bit hard to follow, the quality of the writing could overall be improved.

The presented method detects skull fractures only in 2D while the scans are 3D. The evaluation therefore also did not consider how well the detection results of neighboring 2D slices agree in 3D.

A severe limitation seems to be the way the training/test data is split - it appears as if the 100 test slices were just randomly sampled from all annotated slices. It is not clear whether therefore neighboring, or almost neighboring, slices from the same patients were in the training and the test sets. Especially since fractures will often be visible in multiple slices, this would be a severe issue of the evaluation.

**Detailed Comments:**

In Figure 1, it would be helpful to explain the colors in the figure caption.

Figure 2 would be more clear if there was another path from the classification layer for the case that a proposal is rejected ("not a fracture"). The proposal layer should be called "non-maximum suppression layer", there is nothing else happening in this step, right? I also didn't get why there are three arrows from the feature network to ROI align, what does that indicate (compared with just a single arrow like everywhere else)?

Table 2 has the wrong caption.

The authors draw the conclusion that Skull R-CNN works better for skull fracture detection than Faster R-CNN, but it seems a bit questionable to me whether it is possible to draw this conclusion. The amount of training and test data is very limited, especially compared with the amounts of data Faster R-CNN is typically trained with. The conclusions could therefore be a bit more moderate.

**Justification Of Rating:**

This is a somewhat hard to read paper with limitations in the experimental setup, but with at its core a small but interesting idea. If the authors are able to incorporate some of the feedback, it might make for a decent contribution to MIDL 2020.

**Paper Type:**

methodological development

**Special Issue:**

no

---

> ### Author Response · Authors · 2020-03-24
> **The figures and tables will be revised and the dataset have been reset.**
>
> We are grateful to the reviewer for their thorough assessment of our work and their precise remarks.
>
> -The presented method detects skull fractures only in 2D while the scans are 3D. The evaluation therefore also did not consider how well the detection results of neighboring 2D slices agree in 3D.
>
> Re: Compared to the 3D data, the totoal number of the 2D samples will be much larger. And on the other hand, 3D CNNs have more complex structure than 2D, which will make the training process more difficult. So for our existing data, 2D model is the better choice. In our future work, as the collected data inccreasing, we will consider to use the 3D model or multiple 2D CNNS to improve the performance.
>
> -The evaluation therefore also did not consider how well the detection results of neighboring 2D slices agree in 3D.
>
> Re: Thanks for this helpful comment. We will add the subjective results of the neighboring results in the revised manuscript.
>
>
>
> - A severe limitation seems to be the way the training/test data is split - it appears as if the 100 test slices were just randomly sampled from all annotated slices. It is not clear whether therefore neighboring, or almost neighboring, slices from the same patients were in the training and the test sets. Especially since fractures will often be visible in multiple slices, this would be a severe issue of the evaluation.
>
> Re: We agree that this is a very helpful comment. We reset the dataset, 25 scans with 491 slices are selected as the training set, 10 scans with 208 slices as the validation set, 10 scans with 173 slices as the test set. And the new results show that the Faster R-CNN get the AP of 0.557, and 0.542 on the validation set and test set; Skull R-CNN get the AP of 0.651 and 0.600. All these results, including the PR cures and the subjective results, will be updated in the final version.
>
>
> -In Figure 1, it would be helpful to explain the colors in the figure caption.Table 2 has the wrong caption.
>
> Re: Thansk for the useful comment. We will revise all of these in the final version.
>
>
> -Figure 2 would be more clear if there was another path from the classification layer for the case that a proposal is rejected ("not a fracture"). The proposal layer should be called "non-maximum suppression layer", there is nothing else happening in this step, right? I also didn't get why there are three arrows from the feature network to ROI align, what does that indicate (compared with just a single arrow like everywhere else)?
>
>
> Re: It is right that the main process in the proposal layer is the nms, and after the nms, we limit the output boxes numbers to match all the batches.
> The three arrows represent three sizes of feature maps, thanks for this comment and we will add the explanation on Figure 2.
>
>
> -The authors draw the conclusion that Skull R-CNN works better for skull fracture detection than Faster R-CNN, but it seems a bit questionable to me whether it is possible to draw this conclusion. The amount of training and test data is very limited, especially compared with the amounts of data Faster R-CNN is typically trained with. The conclusions could therefore be a bit more moderate.
>
> Re: Thanks for the constructive comment. We will claim the experimental situation before we make the conclusion in the final version.

---

### Official Review · AnonReviewer2 · 2020-03-12
**Poorly drafted paper proposing a modified R-CNN for skull fracture detection**

**Rating:** 1
**Confidence:** 5

**Summary:**

The authors propose to detect skull fracture using a modified R-CNN approach. In the proposed architecture, the candidate boxes are computed from a simple skeletonization procedure instead of an RPN. Simultaneously, U-Net architecture is used to extract features. The proposed method is evaluated on CT scans from 45 head trauma patients and compared with the manual annotations by the radiologist.  They obtain an AP of 0.68.

**Strengths:**

1) The idea of generating candidate boxes using skeletonization is interesting.
2) Utilized encoder-decoder architecture to extract features from the skull fracture images. This feature extraction module used along with boxes improves the accuracy of detection.

**Weaknesses:**

1) The paper is poorly written with several grammatical errors and missing sentences. For instance, page 5, first line " Due to that, the candidate boxes......" . It is not clear what is the author is referring to by the phrase "Due to that...". Page 7, the first line "average precise (AP)..." should be corrected to "average precision"

2) The equations are not explained thoroughly. For instance, in equations 3 and 4, what is the need for an exponential (e) for computing the predicted box width and height? .  In equation 7, what are Lc and Lr? What is the reason for weighing Lr by 10?



**Justification Of Rating:**

The paper is very poorly written and very difficult to follow. Besides, few critical equations lack a proper explanation/justification.

The authors claim that the full resolution feature network extracts small features. However, this claim is neither justified theoritically or experimentally.

**Paper Type:**

methodological development

**Special Issue:**

no

---

> ### Author Response · Authors · 2020-03-24
> **The grammar and the formulas have been checked and more comparison expariments have been done**
>
> We thank the reviewer for the valuable feedbacks. We would like to clarify some raised issues.
>
> -page 5, first line " Due to that, the candidate boxes......" . It is not clear what is the author is referring to by the phrase "Due to that...".
>
> Re: we are very sorry for the confusion caused by our statement. Here, what we really want to express is that: Because the candidate boxes are generated based on the origin image, coordinates of parts of the boxes mapped on the feature maps will not be integers. To get more suitable features for every candidate boxes, we choose the ROI-Align proposed in Mask R-CNN to replace the ROI-Pooling. And we will avoid this kind of ambiguity in the final version.
>
>
> -in equations 3 and 4, what is the need for an exponential (e) for computing the predicted box width and height?
>
> Re: Before we listed these formulas, we stated that the form of offsets followed the Faster R-CNN. The variables have some different descriptions, for instance,$w_0=W * e^{\delta_3}$   is corresponding to the $t_w^* =log(w^*/w_a)$  in Faster R-CNN. The purpose is to limit the range of the output and speed up training. These explanations will be added in the final version.
>
>
> -In equation 7, what are Lc and Lr? What is the reason for weighing Lr by 10?
>
> Re: The Lc is the loss of the classification task and Lr is the loss of the regression task. And weighing Lr by 10 is to balance the two losses, this has been proved useful in Faster R-CNN. We agree this will clarify the formulation better and we will include this in the revised manuscript.
>
>
> -The authors claim that the full resolution feature network extracts small features. However, this claim is neither justified theoritically or experimentally.
>
>
> Re: we agree that it is very useful to add the experimental proof. To verify the performance of the proposed full resolution network. To reduce the other influence factors, we designed a new combination Skull R-CNN+FPN to make a comparison to the Skull R-CNN. The Skull R-CNN+FPN gets the AP of 0.626 and 0.579 on the validation set and test set, compared to the origin Skull R-CNN decrease by 0.025 and 0.021. And We collected the slices with the ground truth boxes smaller than 16*16 in validation set and test set. The detection results on these two sets show that , the origin SKull R-CNN improve the AP of 0.026 and 0.047. All these results will be updated in the final version.

---

### Official Review · AnonReviewer3 · 2020-03-13
**A somewhat premature paper on skull fracture detection**

**Rating:** 3
**Confidence:** 4
**Recommendation:** Poster

**Summary:**

A CNN based approach for skull fracture detection in axial 2D slices is presented. The approach is based on the Faster R-CNN in combination with a skul skeleton region proposal. The algorithm results in bonding boxes containing the detected fractures. The method is trained and evaluated on 45 head trauma patients with in total 872 slices with a fixed portion used for training testing and validation. The proposed method is compared to just using the Faster R-CNN. The achieved precision is 0.65.

**Strengths:**

The motivation of the paper is clear and the problem addressed is clinically ´´important and technically difficult. The approach is more or less clearly described and compared to a direct application of the Faster R-CNN network.

**Weaknesses:**

Fractures are typically seen in some orientation better than in others. A 3D approach would be would pose the least limitations on fracture orientation, if a 2D approach is used, the authors should discuss why the application to axial slices is sufficient.
Regarding the proposed method, it is not clear to me, why the multi-scale feature map is created for the whole image but later only used for the patches selected by the region proposal step. It is also not completely clear to my why the Roi-align step is necessary.
The evaluation should not only present the achieved precision but also sensitivity and specificity.

**Justification Of Rating:**

In general the paper is of interets for the reader since and important and diffcult problem is addressed with some success. But there are also severe limitations in clarity and evaluation. Therefore a week accept is recommended

**Paper Type:**

validation/application paper

**Questions To Address In The Rebuttal:**

- Would the sensitivity of the method increase if applied in addition to sagittal and coronal slices?
- The seperation between training, vaidation and test sets should make sure that slices from one patient are not distributed over different sets. Was that the case?

**Special Issue:**

no

---

> ### Author Response · Authors · 2020-03-24
> **The datasets have been reset to make sure the slices from one patient  are distributed in only one set.**
>
> We thank the reviewer for the valuable and positive feedbacks. We would like to clarify some raised issues.
>
> -Fractures are typically seen in some orientation better than in others. A 3D approach would be would pose the least limitations on fracture orientation, if a 2D approach is used, the authors should discuss why the application to axial slices is sufficient.
>
> Re: we agree that 3D CNNs can better capture the spatial information than 2D models. But compared to the 3D data, the total number of the 2D samples will be much larger. And on the other hand, 3D CNNs have more complex structure than 2D, which will make the training process more difficult. So for our existing data, 2D model is the better choice. In our future work, as the collected data increasing, we will consider to use the 3D model or multiple 2D CNNS to improve the performance.
>
>
> -Would the sensitivity of the method increase if applied in addition to sagittal and coronal slices.
>
> Re:  We believe that if considering the results of the three directions synthetically, the performance will be better.
>
>
> -why the multi-scale feature map is created for the whole image but later only used for the patches selected by the region proposal step. It is also not completely clear to my why the Roi-align step is necessary.
>
> Re: The skull R-CNN is a kind of the region proposal model, the first step is generated many candidate boxes with different sizes to contain the objects. And the smaller objects need more local information and the lager object need larger receptive field. So the smaller boxes need the feature maps with the larger resolution. As the Figure2 shows, three kinds of feature maps are created for the different sizes of candidate boxes.
>  To score the boxes by the same model, all the regions of the boxed mapped on the feature map will be divided to the same size, like 5*5. But some divided coordinates are not integers,  ROI-Pooling just use the adjacent values, ROI-Align use the interpolation to get the accuratevalues of these points.
>
>
> -The seperation between training, vaidation and test sets should make sure that slices from one patient are not distributed over different sets. Was that the case?
>
> Re: We agree that this comment is very helpful. We reset the dataset, 25 scans with 491 slices are selected as the training set, 10 scans with 208 slices as the validation set , 10 scans with 173 slices as the test set. And the new results show that the Faster R-CNN get the AP of 0.557, and 0.542 on the validation set and test set; Skull R-CNN get the AP of 0.651 and 0.600. All these results, including the PR cures and the subjective results, will be updated in the final version.
>
>
> -The evaluation should not only present the achieved precision but also sensitivity and specificity.
>
> Re: Re: Each of the candidate boxes have a score, but this score is not like that the boxes have the score overall 0.5 are the true boxes. For the different threshold of the score, a boxes could have different classification result, so there will be different TP,FN... and different sensitivity. These performance can be read in the PR cures in figure5, lower the threshold of the score is, the recall is higher and the precision is lower. In the test set, when the recall is about 0.7, the precision is about 0.6.

---

> > ### Comment · AnonReviewer3 · 2020-04-03
> > **some confusion remains**
> >
> > Dear authors, thank you for you comments.
> > Actually, your first comment did not make it much clearer for me. An what do you mean by  "considering the results of the three directions synthetically"?
> > Thanks for the data set separation and for the extend presentation of results.

---

> > > ### Author Response · Authors · 2020-04-04
> > > **Futher explanation**
> > >
> > > -your first comment did not make it much clearer for me
> > >
> > > Re: The inputs of the 3D model will be a 3D data with the size of  512*512*N. It is obvious that N could not be a small number, or the 3D data will have no much more information than 2D. For example, we have 45 CT scans now, if N is set as 32, finally no more than 150 samples will be obtained, and due to the distribution of the fractures, there are many samples do not have a fracture. On the contrary,  we obtained more than 800 2D samples in total, and all of them have at least one fracture. On the other hand, the parameters of a 3D model are much more than the a 2D model, more parameters mean that the model is more difficult to be trained to convergence, especially on the limit dataset. On this kind of small dataset, 2D model can better explore the information than 3D model.
> > >
> > >
> > > -what do you mean by  "considering the results of the three directions synthetically"?
> > >
> > > Re:We believe that, if we get the detectation results on the axial, sagittal and coronal slices seperately,  and combine these results to obtain a final result, the final result will be more accurate. Thanks for your heuristic comment again, this will be a important part in our futher study.

---

### Official Review · AnonReviewer4 · 2020-03-13
**It is not clear if the improvements are from detecting smaller regions better or not**

**Rating:** 2
**Confidence:** 3

**Summary:**

The authors developed a skull fracture detection model based on faster R-CNN. The aim is to better detect fractures in small regions and reduce false positives. U-net based full resolution feature extraction network was combined with skeleton-based region proposals to achieve the aim.  Detecting small regions of interest automatically particularly for skull fractures is significant.

**Strengths:**

The combination of the non-learning based skeletonization method with the learning-based CNN feature extractor is valuable. The use of a CNN with full resolution feature extraction is aimed to detect smaller objects.

**Weaknesses:**

Data splitting is problematic. 2D slices from a subject can be randomly assigned to train/validation/test sets. Subject-based splitting is recommended.

Experiments were not performed to prove the proposed network is indeed improving the detection by detecting smaller regions or due to other reasons. Table 2 should be extended to include findings on only small regions and further experiments should be performed to verify the main hypothesis.




**Detailed Comments:**

Why did the authors use axial images with 2D CNNs only? Why not use 3D CNNs or multiple 2D CNNs to incorporate information from the volume for the task?

Please experiment with the benefit of using the skeletonization method in the task. How much improvement will be lost if you don't use such an approach but identify the coordinates by learning as well?

Add AP values on figures 5 and 6


**Justification Of Rating:**

The paper proposes a method to better detect small fracture regions, however, it is not clear if the proposed network does actually detects small regions or not. There are issues with data split as well.

**Paper Type:**

methodological development

**Special Issue:**

no

---

> ### Author Response · Authors · 2020-03-24
> **The datasets have been reset and the more related experiments have been done.**
>
> We thank the reviewer for the valuable feedbacks. We would like to clarify some raised issues.
>
> -Why did the authors use axial images with 2D CNNs only? Why not use 3D CNNs or multiple 2D CNNs to incorporate information from the volume for the task?
>
> Re: we agree that 3D CNNs can better capture the spatial information than 2D models. But compared to the 3D data, the total number of the 2D samples will be much larger. And on the other hand, 3D CNNs have more complex structure than 2D, which will make the training process more difficult. So for our existing data, 2D model is the better choice. And in our future work, as the collected data increasing, we will consider to use the 3D model or multiple 2D CNNS to improve the performance.
>
>
> -Add AP values on figures 5 and 6
>
> Re: Thanks for the helpful suggestions, we will update these in the revised manuscript.
>
>
> -Data splitting is problematic. 2D slices from a subject can be randomly assigned to train/validation/test sets. Subject-based splitting is recommended.
>
> Re: We agree that subject-based splitting is more reasonable. We reset the dataset, 25 scans with 491 slices are selected as the training set, 10 scans with 208 slices as the validation set, 10 scans with 173 slices as the test set. And the new results show that the Faster R-CNN gets the AP of 0.557, and 0.542 on the validation set and test set; Skull R-CNN gets the AP of 0.651 and 0.600.  All these results, including the PR cures and the subjective results, will be updated in the final version.
>
>
> -Table 2 should be extended to include findings on only small regions and further experiments should be performed to verify the main hypothesis.
>
> Re: Thanks for this useful suggestion. We collected the slices with the ground truth boxes smaller than 16*16, 101 slices of the validation set and 93 slices of the test set were obtained. Finally, Faster R-CNN gets the AP of 0.594 and 0.493, while Skull R-CNN gets 0.674 and 0.633. The improvement is more obvious.
>
> -Please experiment with the benefit of using the skeletonization method in the task. How much improvement will be lost if you don't use such an approach but identify the coordinates by learning as well?
>
> Re: Thanks for the constructive comment. To reduce the other influence factors, we designed a new combination Skull R-CNN+FPN to make a comparison to the Faster R-CNN + FPN. The Skull R-CNN+FPN gets the AP of 0.626 and 0.579 on the validation set and test set.   Compared to the Faster R-CNN + FPN, with the help of the  skeletonization method, the Skull R-CNN+FPN improves the AP of 0.069 and 0.037 on the two sets.  And for the small objects, the AP values are 0.647 and 0.586.  These experiment results will be updated in the final version. A simple table of the AP values can be seen as following.
>
>                                               val         test              val(small objects)      test(small objects)
> Faster R-CNN+FPN           0.557     0.542                 0.594                                 0.493
> Skull R-CNN+FPN              0.626    0.579                  0.647                                 0.586
> Skull R-CNN                       0.651     0.600                 0.673                                 0.633

---

### Meta-Review · Area_Chair1 · 2020-04-04
**MetaReview of Paper177 by AreaChair1**

**Rating:** 3
**Recommendation For Accepted Papers:** Poster

**Metareview:**

I agree with the reviewers that the paper is interesting and that the addressed problem is clinically ´´important and technically difficult. The authors have performed thorough revision and addressed issues raised by the reviewers. I appreciate that the authors changed the experimental settings to address the major issue of data separation and provided updated results.
As pointed by the reviewers, the paper contains numerous language issues that hamper the reading and understanding of the work. Hence, I strongly advise the authors to very carefully check the language.


**Paper Type:**

methodological development

**Special Issue:**

no

---

> ### Author Response · Authors · 2020-04-06
> **Response to the MetaReview**
>
> Thank you for your time and a positive review on our paper.
>
> We will carefully check the language to promise the camera-ready version easy to read.

---

### Decision · Program_Chairs · 2020-04-11

Accept